# Brain activity response to cues during gait in Parkinson's disease: A study protocol

**Rodrigo Vitório**[1], **Rosie Morris**[1], **Julia Das**[1], **Richard Walker**[2], **Martina Mancini**[3], **Samuel Stuart**[1]*

**1** Department of Sport, Exercise and Rehabilitation, Northumbria University, Newcastle upon Tyne, United Kingdom, **2** Northumbria Healthcare NHS Foundation Trust, North Tyneside General Hospital, Newcastle upon Tyne, United Kingdom, **3** Department of Neurology, Oregon Health and Science University, Portland, Oregon, United States of America

* sam.stuart@northumbria.ac.uk

**Data Availability Statement:** No datasets were generated or analysed during the current study. All relevant data from this study will be made available upon study completion.

## Abstract

Various cueing strategies (internal and external) have been used to alleviate gait deficits in Parkinson's disease (PD). However, it remains unclear which type of cueing strategy is most effective at different disease stages or with more severe walking impairment, such as freezing of gait (FOG). The underlying neural mechanisms of response to cueing are also unknown. This trial aims to: (i) determine brain activity response to cue stimulus (internal, visual, auditory or tactile) when walking in PD and; (ii) examine changes in brain activity to cues at different stages of PD. This ongoing single-site study uses an exploratory observational design, with laboratory application of cues for gait deficit. A total of 80 people with PD who meet the inclusion criteria will be enrolled. Participants are split into groups dependent on their disease stage (classified with the Hoehn and Yahr (H&Y) scale); n = 20 H&YI; n = 30 H&YII; n = 30 H&YIII. Within the H&Y stage II and III groups, we will also ensure recruitment of a sub-group of 15 individuals with FOG within each group. Participants perform walking tasks under several conditions: baseline walking without cues; randomized cued walking conditions [internal and external (visual, auditory and tactile) cues]. A combined functional near-infrared spectroscopy and electroencephalography system quantifies cortical brain activity while walking. Inertial sensors are used to assess gait. Primary outcome measures are cue-related changes in cortical brain activity while walking, including the relative change in cortical $HbO_2$ and the power spectral densities at alpha (8-13Hz), beta (13-30Hz), delta (0.5-4Hz), theta (4-8Hz) and gamma (30-40Hz) frequency bandwidths. Secondary outcome measures are cue-related changes in spatiotemporal gait characteristics. Findings will enhance our understanding about the cortical responses to different cueing strategies and how they are influenced by PD progression and FOG status. This trial is registered at clinicaltrials.gov (NCT04863560; April 28, 2021, https://clinicaltrials.gov/ct2/show/NCT04863560).

## Introduction

Gait impairments are a common and early feature of Parkinson's disease (PD) [1], and are a major cause of functional dependence, falls and death [2]. Gait impairments in PD can be

**Funding:** This study has been funded by Parkinson's Foundation (PI: Dr Samuel Stuart). Dr Stuart is supported, in part, by a Parkinson's Foundation Post-doctoral Fellowship for Basic Scientists (PF-FBS-1898-18-21) and a Clinical Research Award (PF-CRA-2073).

**Competing interests:** The authors have declared that no competing interests exist.

continuous or intermittent, with deterioration of gait with disease progression. Continuous impairments of gait in PD include reduced step length, speed, increased double-support time and arrhythmicity [1, 3]. Intermittent gait deficits include freezing of gait (FOG) [3], which involves a failure to initiate or maintain walking. FOG is particularly evident in later disease stages (Hoehn & Yahr (H&Y) stages III-V) [4], and are frequently elicited during turning, gait initiation or with greater task complexity such as walking through doorways or navigating obstacles [5]. Within PD there is a shift from automatic to more conscious control of gait [6, 7], with increased executive-attentional deployment required to overcome deficits in subcortical regions, which is thought to be exacerbated in people with FOG [8]. Gait deficits in PD have largely been attributed to reduced dopamine within the nigro-striatal pathway [9], but this does not fully explain gait changes in PD across disease severity [10].

There is currently no definitive treatment for gait impairment in PD. Dopaminergic medication has limited effect on gait deficits in PD [10, 11]. Despite improvement in selective characteristics, i.e. speed and step length largely improve, deficits in gait do not resolve completely compared to controls with levodopa replacement [12]. Similarly, recent cholinergic therapy shows promise [13–15] but despite this current evidence for pharmacological intervention for gait impairment remains limited. Deep brain stimulation also has limited impact on gait deficits [16] and can sometimes have a negative effect [17]. Clinical implementation of non-pharmacological therapies, such as cueing, can help to ameliorate gait issues and reduce falls risk [18–21]. Internal (i.e. thinking about larger steps) and external cue stimuli (e.g. auditory (i.e. metronome beats to step in time with), visual (i.e. transverse taped lines on the floor to step on or over) or tactile (i.e. vibration feedback to step in time with) cues) are recommended therapy strategies for Parkinsonian gait impairment [22], which are easily implemented within the home by patients, their family members/carers or clinicians.

Physical therapy guidelines suggest that different cueing strategies (internal or external) may be useful for gait improvement at specific PD stages [23] (e.g. H&Y I-III). However, cue recommendations for PD are vague (i.e.. do not thoroughly differentiate cue effectiveness at PD stages), are based on subjective clinical judgement and are supported by limited objective evidence. Previous cueing studies have largely been conducted in mid-late stage (H&Y II-IV) PD participants. Such studies suggest that in the later stages of PD, when FOG and cognitive deficits may be prevalent, only external cues are effective [24, 25]. Whereas internal cues are thought to be most useful in early-stage disease (H&Y I) when cognition is largely intact. Alternatively, recent evidence has suggested that internal cues may be more effective at improving gait than external cues in mid-late stage PD (H&Y II-III) [26], which directly contrasts previous physical therapy recommendations. Furthermore, evidence for early PD stage cueing is lacking. A recent longitudinal pilot study in nine PD participants demonstrated that auditory cue response may alter with PD progression, with limited gait benefits or possible cue induced impairments (e.g. increased variability) in early stage PD (within 6 months of diagnosis: H&Y I-II) but increased gait benefits at later stage PD (3 years post diagnosis: H&Y II-III) [27]. The neural mechanisms underlying cue response in PD are poorly understood, which has led to a lack of knowledge related to cue implementation stage and long-term cue use, as well as reports of variable response [28] and selective short-term gait characteristic improvement [29].

Internal and external cue stimulus can improve gait in PD, but response is poorly understood, although attentional mechanisms may be key. Theories suggest that different cueing strategies may be underpinned by different attentional mechanisms. External auditory cues are suggested to improve gait through replacement of rhythmic basal ganglia output via external stimuli that reduces prefrontal cortex (PFC) burden during gait [30], and visual or tactile cues may activate attentional mechanisms [19] to allow faster subconscious processing of

sensory information when walking [31, 32]. Whereas, internal strategies may use attentional mechanisms that by-pass defective basal ganglia circuitry to improve gait [33]. To date however, these theories remain relatively unexplored, likely due to an inability to image the brain during walking. Therefore, specific cortical contributions to different cueing strategies remain unknown and warrant further investigation to inform the most effective strategy to alleviate gait deficit in people with PD.

This is the first study to investigate comprehensive brain activity during walking, and response to internal and external cue stimulus, across different PD stages. Recent technological developments have allowed monitoring of brain activity during real-time gait. Progression in mobile brain imaging techniques, such as non-invasive mobile functional near-infrared spectroscopy (fNIRS) and electroencephalography (EEG) head caps, have allowed cortical brain activity to be monitored in PD during actual gait, but response to various cueing stimuli across disease stages has yet to be fully determined. Our recent systematic review demonstrated that there is increased activation of specific cortical regions when walking in PD compared to controls, particularly at the PFC [34]. Of note, the majority of previous studies have largely been limited to measurement of only PFC activity due to technology constraints (i.e. fNIRS headbands over the forehead). Our recent fNIRS study showed no change in PFC activity in PD with tactile cues compared to usual walking, which was despite gait improvements with cueing [35]. Although people with FOG have increased PFC activity during usual walking relative to those without FOG [8], both groups responded similarly to cueing [35]. However, this study was limited to the PFC only. Additionally, EEG studies have shown that those with FOG have increased supplementary motor area and frontal cortex activity during FOG episodes when walking, but studies have largely been limited to single EEG channel analysis (i.e. Pz, Cz, Fz or Oz) [34], which limits interpretation of specific brain regions. Therefore, brain regional analysis is vital, with previous studies highlighting that cognitive function is associated with selective gait outcomes in PD [36]. For example, attention (involving the PFC) has been related to control of pace-related gait characteristics such as step length and speed, whereas visuo-spatial ability (involving the parietal cortex) has been associated with variability and timing gait characteristics [36].

### Aims and hypotheses

**Aim I.** Determine brain activity response to cue stimulus (internal, visual, auditory or tactile) when walking in people with PD with and without FOG. We hypothesize that internal cueing will demand more PFC activity compared to usual walking and external stimulus strategies, with additional sensorimotor integration required with external cues in those with FOG. Additionally, we hypothesize that external auditory, visual and tactile cueing will be underpinned by selective cognitive, motor and sensorimotor regional brain activation, with more diffuse activation in those with FOG.

**Aim II.** Examine changes in brain activity to cues at different stages of PD. We hypothesize that H&YI will respond more to internal than external cue stimulus, but with the least activation of the PFC at this PD stage compared to H&YII and H&YIII. Additionally, we hypothesize that external cue stimulus will elicit multi-region cortical response at H&YIII, with activation of PFC, motor and parietal regions dependent upon cue stimulus type.

## Materials and methods

### Study design

This study uses an exploratory observational design, with laboratory application of cues for gait deficit. The trial compares cortical brain activity response to multiple cueing strategies

| | STUDY PERIOD | |
| --- | --- | --- |
| | **Enrolment** | **Post-allocation** |
| **TIMEPOINT** | *entry* | *Day1* |
| **ENROLMENT:** | | |
| **Eligibility screen** | X | |
| **Informed consent** | X | |
| **INTERVENTIONS:** | | |
| *Internal cues* | | X |
| *External cues* | | X |
| **ASSESSMENTS:** | | |
| *Inclusion/exclusion criteria* | X | |
| *Clinical assessment* | | X |
| *Gait and brain activity* | | X |

**Fig 1. SPIRIT template for the schedule of enrolment, interventions, and assessments.**

(internal, visual, auditory and tactile) when walking. SPIRIT template for the schedule of enrolment, interventions, and assessments is presented in Fig 1. The trial design is illustrated in Fig 2. Recruitment began in May 2021. People with PD willing to participate in the study are screened for eligibility and, once confirmed, undergo clinical and gait (with and without cues) assessments while ON medication in a single study visit of approximately 2.5 hours. This protocol paper follows the SPIRIT (Standard Protocol Items: Recommendations for Interventional Trials) 2013 statement and guidelines [22].

**Fig 2. Flowchart of the study.**

## Study setting

This trial is carried out at the Clinical Gait Laboratory at Coach Lane Campus, Northumbria University, Newcastle upon Tyne.

## Ethical approval and registration

This study conforms to the Declaration of Helsinki and has been approved by the London-Bloomsbury NHS Research Ethics Committee (and Health Research Authority; 20/LO/1036, 05/10/2020). Participation in the study is voluntary and requires written informed consent from each participant, which will be obtained by the principal investigator or an approved team member. This trial is registered at clinicaltrials.gov (NCT04863560; 28 April 2021; https://clinicaltrials.gov/ct2/show/NCT04863560).

## Recruitment and eligibility criteria

Participants are recruited from Movement Disorder Clinics at Northumbria Healthcare NHS Foundation Trust and through the Parkinson's UK research excellence network (i.e., advertisement on website and newsletters) and the DeNDRoN Research Case Register. Patients with PD are informed about the study and those who agree to learn more about it are contacted by a member of the research team who explains the study further. If sufficiently interested, patients are invited and given a Participant Information Sheet concerning the study. The invitation is followed up by a telephone call to assess willingness to participate. If willing, a mutually convenient time for the study visit is organised.

Inclusion criteria are the following: Clinical diagnosis of PD by a movement disorder specialist according to UK brain bank criteria; H&Y stage I-III; aged >50 years; able to walk and stand unaided; adequate hearing (as evaluated by the whisper test; stand 2 metres behind subject and whisper a 2 syllable word, subject repeats word) and vision capabilities (as measured using a Snellen chart—6/18-6/12); stable medication for the past 1 month and anticipated over a period of 6 months.

Exclusion criteria are: Psychiatric co-morbidity (e.g. Schizophrenia, major depressive disorder as determined by geriatric depression scale—GDS-15; <10 [37]); clinical diagnosis of dementia or other severe cognitive impairment (Montreal cognitive assessment <21 [38]); history of neurological disorders other than PD (e.g. Huntington's disease, stroke, traumatic brain injury, multiple sclerosis, Alzheimer's disease etc.); acute lower back or lower extremity pain, peripheral neuropathy, rheumatic and orthopedic diseases; unstable medical condition including cardiovascular issues (e.g. angina, myocardial infarction, pulmonary embolism etc.) in the past 6 months; unable to comply with the testing protocol; and interfering research project or clinical therapy (e.g. currently involved in another clinical trial at the hospital or university involving pharmaceuticals, exercise or any other intervention that may impact their ability to walk).

## Assignment and blinding

A total of 80 participants will be enrolled in the study. Participants are split into groups dependent on the stage of their disease (classified with H&Y scale); n = 20 H&YI (early disease, minimal symptoms); n = 30 H&YII (mild disease, no balance issues); n = 30 H&YIII (moderate disease, balance issues). Within the H&Y stage II and III groups, we will also ensure recruitment of a sub-group of n = 15 individuals who self-report FOG within each group (n = 30 total with FOG), which will provide a sub-group for further data analysis. We limit FOG sub-group recruitment to these groups as we do not expect any individuals with FOG to be in H&YI. Self-reported FOG is based upon a question in the new Freezing of Gait Questionnaire after seeing

the short clip related to the questionnaire [39]. Participants are categorized as "freezers" if they have experienced such a feeling or episode over the past month. In addition, we evaluate the presence of FOG in the laboratory during clinical examination, and if patients are seen with FOG but report 0 on the FOG questionnaire, they are considered freezers.

Blinding is not possible for participants or assessors due to the use of specific equipment and/or instructions for the application of different cue strategies while walking. To minimize the risk of bias, recruitment and consent form do not mention potential superior effectiveness of a given cue strategy and instructions for application of cues are standardized.

## Clinical assessment

Initially, participants undergo a clinical assessment, which includes a structured anamnesis (socio-demographic information and medical history) and cognitive and clinical tests. Motor signs related to PD severity are assessed with the motor section of the Movement Disorder Society Unified Parkinson's Disease Rating Scale [40]. Global cognition is assessed with the Montreal Cognitive Assessment [41]. Attention is measured with a computerized button pressing test, involving simple and choice reaction time, and digit vigilance. Executive function is measured using Royall's clock drawing [42] and Trail-making Test Part B-A. Working memory and visuo-spatial ability are measured through seated forward digit span and judgement of line orientation tasks [43], respectively. Basic visual functions of visual acuity and contrast sensitivity are assessed using standardized charts (logMar and logCS). Fear of falling will be measured using the falls efficacy scale–international version [44].

## Gait assessment and interventions

After the clinical assessment, participants perform walking tasks on a 9-m travel path under several conditions while instrumented with an fNIRS/EEG head cap and 8 inertial sensors. A baseline walking condition (i.e., walking at self-selected comfortable pace without cues) is performed first, followed by randomized cued walking conditions. Internal cue: participants are instructed to think about taking bigger steps while walking. External visual cues: transverse tapes (3 cm wide) are placed on the travel path and the distance between tapes is set at the individual step length (as measured during baseline walking); participants are instructed to step over the tapes. External auditory cues: participants are instructed to step in time to the beats of an electronic metronome set at their baseline cadence. External tactile cues: participants are instructed to step in time to metronome-like vibrations (also set at their baseline cadence) provided on their wrists trough a pair of bracelets (Pulse, Soundbrenner). For all walking conditions, participants stand still for 20 seconds before and after the walk bout, which lasts 80 seconds.

## fNIRS/EEG: Cortical brain activity

This study uses a non-invasive and mobile combined fNIRS (OctaMon + Brite 24, Artinis Medical Systems, The Netherlands) and EEG system (SAGA 32-channel, TMSi, The Netherlands) to record cortical brain activity while walking. The fNIRS/EEG head cap covers and records signals from the frontal, central, parietal and occipital brain regions. The head cap includes 32 electrodes and 28 optodes (consisting of 18 transmitters and 10 detectors). The fNIRS and EEG signals are synchronized and recorded at 50Hz (Oxysoft) and 1000Hz (SAGA Data Recorder 32+), respectively. Data analysis for fNIRS and EEG signals is conducted separately, but we will compare outcomes from simultaneous recordings from the two systems, which will ensure robust results.

fNIRS data analysis is performed in line with our previous reliable fNIRS walking studies in PD [8, 35, 45]; 1) oxygenated (HbO$_2$) and deoxygenated (HHb) hemoglobin signals are low-

pass filtered (cut-off 0.14Hz); 2) Corrected for baseline (removing median of 20 seconds of initial standing before walks from signal); 3) Reference channel corrected (short 1 cm channel is taken away from long 3 cm channel signals); 4) Visual signal inspection; and 5) Averaging across fNIRS channels for regions of interest (ROI). A 3D digitizer (Polhemus Patriot) is used to obtain morphological locations for cortical ROIs relative to scalp position and the fNIRS optodes. Data from the digitizer is entered into the software package NIRS-statistical package metric mapping (NIRS-SPM, http://www.nitrc.org/projects/nirs_spm), which is implemented in Matlab. NIRS-SPM allows registration of fNIRS channel data onto the Montreal Neurological Institute standard brain space using probabilistic registration of the fNIRS co-ordinate data to determine channels related to ROIs at the group level.

EEG data analysis is performed in line with our previous data analysis. Initial signal processing is conducted using the EEGLAB toolbox within Matlab [46], which involves band-pass filtering (1-250Hz) and extraction of separate brain and artefact sources in the EEG signals with Independent Component Analysis [47]. Source localized independent components (ICs) are derived and ICs related to brain activity are identified using the ICLabel function [48]. ICs are clustered (via dipoles and K-means) according to anatomical location, with power spectral density (PSD) extracted from each cluster.

## Inertial sensors: Gait

Wearable inertial measurement units (Opal, APDM Wearable Technologies, USA) are used to quantify spatiotemporal gait parameters at a sampling rate of 128 Hz. They are located at the sternum and pelvis levels, and bilaterally on the wrists, shanks, and feet of participants. Inertial sensors consist of triaxial accelerometers, gyroscopes, and magnetometers, and are securely fixed to the participant's body with Velcro straps. The inertial sensors and fNIRS/EEG system are synchronized through the Artinis PortaSync. Gait characteristics are extracted from the inertial sensors using the Mobility Lab software, V2 (APDM, USA).

## Outcomes measures

The primary outcome measures for this study are cue-related changes in cortical brain activity during gait measured by the fNIRS/EEG system. Specifically, the outcomes are the relative change in cortical $HbO_2$ and the PSD of the EEG clusters, at alpha (8-13Hz), beta (13-30Hz), delta (0.5-4Hz), theta (4-8Hz) and gamma (30-40Hz) frequency bandwidths.

The secondary outcome measures are cue-related changes in gait characteristics, such as stride length, gait speed, stride time and gait variability (standard deviation of consecutive steps/strides).

## Safety considerations

All measurements and cueing strategies are non-invasive and place participants at no higher risk than those that normally may occur during sitting, standing, or walking. To minimize the risk of falls, a member of the research team walks by the side of the participants to provide support in case they lose their balance. To prevent excessive fatigue, participants are encouraged to take breaks as needed throughout the study procedures.

## Adverse events

Any untoward medical occurrence, injury or any untoward clinical signs in participants, whether or not related to the cueing interventions, will be recorded as an adverse event and managed according to the Health Research Authority (HRA) Guidance.

## Sample size calculation

To determine an appropriate sample size we used our previous fNIRS study that examined response to tactile cueing in PD (n = 25, H&Y I-III) [35]. Given the total effect size of 0.52 for $HbO_2$ difference in walking with and without tactile cues, we require at least 18 people per group (H&YI, II, III) for adequate power (α = 0.05, 1-β = 0.95). Additionally, given the total effect size of 1.43 for mobility differences during tactile cueing between those with (n = 25) and without FOG (n = 18) in our previous study (Mancini et al. 2018), we require at least 14 people with FOG for adequate power (α = 0.05, 1-β = 0.95). Considering that we expect larger effect sizes for more severe people with PD and smaller in less severe people with PD, we should be adequately powered with a total sample size of 80, with groups of ≥20 subjects at each H&Y stage (I-III) and for those with FOG (n = 30 across H&Y stages II and III).

## Statistical analysis

Statistical analysis will be undertaken using SPSS version 25 or more recent versions (SPPS, Inc. an IBM company). All statistical tests will be carried out at the 5% two-sided level of significance. Demographic characteristics and baseline data will be summarized using descriptive statistics, including means, standard deviations, median, minimum, maximum and inter-quartile ranges for continuous or ordinal data and percentages for categorical data. The descriptive statistics will be tabulated and presented graphically for clarity. Outcomes obtained from the walking without cues condition will be taken as baseline values.

One-sample Kolmogorov-Smirnov tests will be used to check for normally distributed data. Non-normally distributed continuous data will be transformed where appropriate to meet the requirements of parametric tests; otherwise, equivalent non-parametric tests will be adopted.

**Hypothesis 1a.** To test whether increased PFC activation occurs with internal compared to external cues in PD, and whether additional sensorimotor integration is required with external cues in FOG, we will use Linear Mixed Effects Models (LMEMs) to compare PFC and parietal $HbO_2$ (fNIRS) and PSD (EEG) values from walking with and without internal and external (auditory, visual, tactile) cues, with FOG status as a between group variable.

**Hypothesis 1b.** To test whether response to external cueing strategies is underpinned by selective cognitive, motor and sensorimotor activity, with more diffuse activity in those with FOG, we will use separate LMEMs to compare regional (entire cortex) $HbO_2$ and PSD values from walking with different external cue stimulus (auditory vs visual vs tactile), with FOG status as a between group variable.

LMEMs will have age and gender as covariates, and Bonferroni correction for multiple comparisons.

**Hypothesis 2a.** To test whether H&YI respond more to internal than external cues with less PFC activity than later PD stages, we will compare PFC $HbO_2$ and PSD data, with and without internal and external cues across PD stages (H&YI vs H&YII vs H&YIII). Separate LMEMs will determine how PFC cue stimulus response changes from H&YI to later stages. FOG status, gender and age will be added as model covariates.

**Hypothesis 2b.** To test if brain activity response to cue stimulus becomes more diffuse in later compared to earlier PD, we will compare regional (entire cortex) $HbO_2$ and PSD values with and without cues across disease stages (H&YI vs II vs III), and within those who do and do not report FOG. Separate LMEMs will determine how brain activity cue stimulus response changes between PD stages, particularly H&YIII compared to H&YI-II. An additional LMEM will examine cue response between those with (n = 30) and without FOG (n = 50). Regression analysis will also be performed between brain activity response (regional HbO2, PSD values) to different cues, disease severity (UPDRS-III) and FOGQ score across the cohort (n = 80).

LMEMs will have age and gender as covariates, and Bonferroni correction for multiple comparisons.

## Data management and availability

The study complies with the General Data Protection Regulation (GDPR) and Data Protection Act 2018, which require data to be de-identified as soon as it is practical to do so. All data samples collected as part of this study are anonymised with participants being assigned a unique study code. We keep one hard copy of the assessment in locked filing cabinets in the Clinical Gait Laboratory, Coach Lane, Northumbria University. This is the only place where we store any personal details. This information is kept locked away and is only available to research staff directly running the study. All data are entered into an electronic database using unique study codes for each participant and are securely stored on a password-protected computer database. Any significant protocol modifications during this study will be communicated to the trial registry. Deidentified research data will be made publicly available (online repository) when the study is completed and published. Results will be disseminated through academic outputs (e.g., national and international conferences and publications), and through hosting continued professional development workshops and/or general public meetings.

## Discussion

Despite the common clinical application of cueing strategies to alleviate gait deficits in PD, little is known about the underlying neural mechanisms of response to cueing. This is the first study to investigate comprehensive cortical responses to different cueing strategies and how they are influenced by PD progression and FOG status. Findings will provide a detailed description of how different cortical regions respond to different cueing strategies in people with PD, accounting for disease stage and FOG status. Findings will also provide evidence to support a more tailored approach for the application of cues to improve gait in PD.

This study has two major strengths, including the concurrent assessment of gait and brain cortical responses to cueing and the comparison of response to various cueing strategies across PD stages. The combination of mobile brain imaging techniques (fNIRS and EEG) with wearable inertial sensors for the walking assessment will provide a more complete understanding of the response to cues in PD, allowing direct associations between underlying cortical mechanisms and cue-related changes in gait. Additionally, the direct comparison of multiple cueing strategies across PD stages will allow the identification of specific cueing strategies that work best for gait improvement at specific PD stages.

Findings have potential to support the development of future research aiming to maximize the response to cueing in PD. Cortical regions responsive to cueing may be targeted by other interventions, such as non-invasive brain stimulation, to enhance benefits from cueing. Thus, the development of combined interventions (e.g., non-invasive brain stimulation + cueing) would be a reasonable next step to be explored. To date, transcranial direct current stimulation has been shown to enhance the effects of other combined interventions, such as exercise or cognitive training, in people with PD [49, 50].

We acknowledge the following limitations, which will be considered for the final interpretation of findings. First, this study is limited by the lack of blinding. Given the nature of clinical application of cueing strategies, which involves equipment and/or specific instructions to patients, we are unable to blind participants or assessors. This methodological aspect may introduce performance bias on findings, resulting in inflated treatment effects. If that is the case, it is very likely that the performance bias will be consistent across the tested cueing strategies because participants are not given information about potential superior effectiveness of a

given cue strategy and standardized instructions are given to participants during the application of cues. Second, because this study is focused on the immediate response to cues (i.e. single exposure), interpretations will not be generalized to long-term application and potential adaptation to cues. These aspects will remain unexplained and should be targeted in future research.

## Supporting information

**S1 File. SPIRIT checklist.**
(DOCX)

**S2 File. Study protocol (Ethics approved).**
(DOCX)

## Acknowledgments

This study is based at the Physiotherapy Innovation Laboratory (Website: www.pi-lab.co.uk, Twitter: @Physio_In_Lab) and authors acknowledge participants for donating their time.

## Author Contributions

**Conceptualization:** Samuel Stuart.

**Data curation:** Rodrigo Vitório, Rosie Morris, Julia Das, Samuel Stuart.

**Formal analysis:** Samuel Stuart.

**Funding acquisition:** Samuel Stuart.

**Investigation:** Martina Mancini, Samuel Stuart.

**Methodology:** Rodrigo Vitório, Rosie Morris, Julia Das, Richard Walker, Martina Mancini, Samuel Stuart.

**Project administration:** Rodrigo Vitório, Rosie Morris, Julia Das, Richard Walker, Samuel Stuart.

**Resources:** Samuel Stuart.

**Supervision:** Richard Walker, Martina Mancini, Samuel Stuart.

**Writing – original draft:** Rodrigo Vitório.

**Writing – review & editing:** Rodrigo Vitório, Rosie Morris, Julia Das, Richard Walker, Martina Mancini, Samuel Stuart.

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
