## [Decision Letter · Decision Letter 0]

1 Jul 2022

PONE-D-21-34386

Brain activity response to cues during gait in Parkinson's disease: a study protocol

PLOS ONE

Dear Dr. Stuart,

Thank you for submitting your manuscript to PLOS ONE. After careful consideration, we feel that it has merit but does not fully meet PLOS ONE’s publication criteria as it currently stands. Therefore, we invite you to submit a revised version of the manuscript that addresses the points raised during the review process.

We look forward to receiving your revised manuscript.

Kind regards,

Parasuraman Padmanabhan, Ph.D

Academic Editor

PLOS ONE

Journal Requirements:

5. We note that the original protocol file you uploaded contains a confidentiality notice indicating that the protocol may not be shared publicly or be published. Please note, however, that the PLOS Editorial Policy requires that the original protocol be published alongside your manuscript in the event of acceptance. Please note that should your paper be accepted, all content including the protocol will be published under the Creative Commons Attribution (CC BY) 4.0 license, which means that it will be freely available online, and any third party is permitted to access, download, copy, distribute, and use these materials in any way, even commercially, with proper attribution.

Therefore, we ask that you please seek permission from the study sponsor or body imposing the restriction on sharing this document to publish this protocol under CC BY 4.0 if your work is accepted. We kindly ask that you upload a formal statement signed by an institutional representative clarifying whether you will be able to comply with this policy. Additionally, please upload a clean copy of the protocol with the confidentiality notice (and any copyrighted institutional logos or signatures) removed.

Reviewers' comments:

Reviewer's Responses to Questions

**Comments to the Author**

1. Does the manuscript provide a valid rationale for the proposed study, with clearly identified and justified research questions?

Reviewer #1: Yes

Reviewer #2: Yes

2. Is the protocol technically sound and planned in a manner that will lead to a meaningful outcome and allow testing the stated hypotheses?

Reviewer #1: Yes

Reviewer #2: Yes

3. Is the methodology feasible and described in sufficient detail to allow the work to be replicable?

Reviewer #1: Yes

Reviewer #2: Yes

4. Have the authors described where all data underlying the findings will be made available when the study is complete?

Reviewer #1: Yes

Reviewer #2: Yes

5. Is the manuscript presented in an intelligible fashion and written in standard English?

Reviewer #1: Yes

Reviewer #2: Yes

6. Review Comments to the Author

You may also provide optional suggestions and comments to authors that they might find helpful in planning their study.

Reviewer #1: The manuscript “Brain activity response to cues during gait in Parkinson's disease: a study protocol” by Stuart et.al. deals with the concurrent assessment of gait and brain cortical responses using different cueing strategies. The research findings will provide support to more tailored approaches for the application of cues to improve gait in Parkinson's disease. These findings should thus motivate further research and would deserve publication.

I am positive about this work and would suggest this work for publication.

Reviewer #2: 1. What type of sampling was adopted in choosing the subjects

2. Statistical analysis of results if possible can be provided, that will improve the impact of the work

3. Hybrid brain imaging modalities are taken, please highlight the cons , while using these modalities individually

4. Future work can be highlighted in the discussion section

5. Recent papers can be included in the introduction session

6. Whether this hybrid medical imaging modality (fNIRs+EEG)is applicable for other neuro disorder analysis

7. PLOS authors have the option to publish the peer review history of their article (what does this mean?). If published, this will include your full peer review and any attached files.

Reviewer #1: No

Reviewer #2: **Yes: **Dr S.N Kumar

---

## [Author Response · Author response to Decision Letter 0]

29 Jul 2022

Dear Dr. Parasuraman Padmanabhan,

Thank you for the opportunity to revise and resubmit the manuscript. Of course, many thanks to the reviewers for taking the time to comment on the manuscript and for providing feedback, which is greatly appreciated. Reviewers’ comments and suggestions helped us to clarify additional aspects in the manuscript. Our specific responses are listed below. 

We have highlighted the changes in the document by using the Track Changes tool. A clean copy of our revised manuscript is also included in the re-submission.

Reviewer #1

The manuscript “Brain activity response to cues during gait in Parkinson's disease: a study protocol” by Stuart et.al. deals with the concurrent assessment of gait and brain cortical responses using different cueing strategies. The research findings will provide support to more tailored approaches for the application of cues to improve gait in Parkinson's disease. These findings should thus motivate further research and would deserve publication. 

I am positive about this work and would suggest this work for publication. 

We appreciate reviewer’s comments and recommendation for publication.

Reviewer #2

1. What type of sampling was adopted in choosing the subjects

No specific type of sampling is adopted. As described in the manuscript, patients with a confirmed diagnosis of Parkinson’s disease are informed about the study and those who agree to learn more about it are contacted by a member of the research team who explains the study further. If sufficiently interested, patients are invited and given a Participant Information Sheet concerning the study. The invitation is followed up by a telephone call to assess willingness to participate. If willing, a mutually convenient time for the study visit is organised.

2. Statistical analysis of results if possible can be provided, that will improve the impact of the work

Please note that this paper is submitted as a “Study Protocol” and, therefore, does not include data/results. A paper describing the observed effects on brain cortical activity and gait parameters is planned for later this year (or early next year). 

3. Hybrid brain imaging modalities are taken, please highlight the cons, while using these modalities individually

EEG and fNIRS quantify different physiological processes related to brain activity. EEG measures the electromagnetic field created when neurons in the brain fire; and fNIRS measures the hemodynamic response, which is the adjustment of oxygen in the blood when a brain region is active. The obvious con of using one of these modalities individually would be the restriction to one physiological process. However, we believe that the discussion of cons regarding the use of a specific brain imaging modality is beyond the scope of our protocol paper and, therefore, we have not added this discussion to the paper. 

4. Future work can be highlighted in the discussion section

Thank you for your suggestion. We have added the following paragraph to the Discussion section: “Findings have potential to support the development of future research aiming to maximize the response to cueing in PD. Cortical regions responsive to cueing may be targeted by other interventions, such as non-invasive brain stimulation, to enhance benefits from cueing. Thus, the development of combined interventions (e.g., non-invasive brain stimulation + cueing) would be a reasonable next step to be explored. To date, transcranial direct current stimulation has been shown to enhance the effects of other combined interventions, such as exercise or cognitive training, in people with PD (47, 48).”

5. Recent papers can be included in the introduction session

Thank you for your suggestion. We have added citations to two recent review papers on cueing in PD: 

- Forte et al., 2021: https://pubmed.ncbi.nlm.nih.gov/34067458/#

- Gonçalves et al., 2021: https://pubmed.ncbi.nlm.nih.gov/33969461/#

6. Whether this hybrid medical imaging modality (fNIRs+EEG)is applicable for other neuro disorder analysis

Thank you for your suggestion. Although we are confident that hybrid fNIRS + EEG is applicable for other clinical populations, this aspect is beyond the scope of our Study Protocol and, therefore, we have not added such information

---

## [Editor Report · Decision Letter 1]

27 Sep 2022

Brain activity response to cues during gait in Parkinson's disease: a study protocol

PONE-D-21-34386R1

Dear Dr. Stuart,

We’re pleased to inform you that your manuscript has been judged scientifically suitable for publication and will be formally accepted for publication once it meets all outstanding technical requirements.

Kind regards,

Parasuraman Padmanabhan, Ph.D

Academic Editor

PLOS ONE
---

## [Editor Report · Acceptance letter]

8 Nov 2022

PONE-D-21-34386R1 

Brain activity response to cues during gait in Parkinson's disease: a study protocol 

Dear Dr. Stuart:

I'm pleased to inform you that your manuscript has been deemed suitable for publication in PLOS ONE. Congratulations! Your manuscript is now with our production department. 

Kind regards, 

on behalf of

Dr. Parasuraman Padmanabhan 

Academic Editor

PLOS ONE